# The Importance of Being PI3K in the RAS Signaling Network

**DOI:** 10.3390/genes12071094

**Published:** 2021-07-19

**Authors:** Cristina Cuesta, Cristina Arévalo-Alameda, Esther Castellano

**Affiliations:** Tumour-Stroma Signalling Laboratory, Centro de Investigación del Cáncer, Instituto de Biología Molecular y Celular del Cáncer, Consejo Superior de Investigaciones Científicas (CSIC)-Universidad de Salamanca, Campus Miguel de Unamuno, 37007 Salamanca, Spain; ccuesta@usal.es (C.C.); crisarevalo@usal.es (C.A.-A.)

**Keywords:** Ras oncogenes, PI3-Kinase

## Abstract

Ras proteins are essential mediators of a multitude of cellular processes, and its deregulation is frequently associated with cancer appearance, progression, and metastasis. Ras-driven cancers are usually aggressive and difficult to treat. Although the recent Food and Drug Administration (FDA) approval of the first Ras G12C inhibitor is an important milestone, only a small percentage of patients will benefit from it. A better understanding of the context in which Ras operates in different tumor types and the outcomes mediated by each effector pathway may help to identify additional strategies and targets to treat Ras-driven tumors. Evidence emerging in recent years suggests that both oncogenic Ras signaling in tumor cells and non-oncogenic Ras signaling in stromal cells play an essential role in cancer. PI3K is one of the main Ras effectors, regulating important cellular processes such as cell viability or resistance to therapy or angiogenesis upon oncogenic Ras activation. In this review, we will summarize recent advances in the understanding of Ras-dependent activation of PI3K both in physiological conditions and cancer, with a focus on how this signaling pathway contributes to the formation of a tumor stroma that promotes tumor cell proliferation, migration, and spread.

## 1. Introduction

Ras proteins are the founding members of the Ras superfamily of GTPases, which in humans is composed of more than 150 members [1,2]. Ras proteins are membrane-bound small GTPases that act as molecular transducers, coupling cell surface receptors to intracellular effector pathways to regulate cellular processes such as cell proliferation, differentiation, migration, and apoptosis [3,4]. In humans, three Ras genes (*H-ras*, *N-ras,* and *K-ras*) encode four distinct Ras proteins: H-Ras, N-Ras, K-Ras4A, and K-RasS4B, the latter 2 resulting from alternative RNA splicing of the *K-ras* gen. These four Ras isoforms are ubiquitously expressed and are highly similar in primary sequence, structure, and biochemical properties [5,6]. They share 90% sequence identity in the G domain [7], being 100% identical in the N-terminus of this domain, termed the effector lobe, and sharing 82% of the sequence of the allosteric lobe [5,8]. In contrast, the C-terminal hypervariable region (HVR) shares little sequence similarity [7,9].

Alterations in Ras signaling are implicated in the development of different diseases, such as neurological disorders, developmental disorders, and autism. Additionally, Ras proteins are recognized as major oncogenes, as mutations in all three Ras genes occur in approximately 30% of human cancers [10]. Mutant Ras is a driver both in tumor initiation and tumor maintenance [5,7]. In general, K-Ras is the most frequently mutated isoform (accounting for 75% of Ras mutation in cancer), followed by N-Ras (17%) and H-Ras (7%) [10]. The frequency in which every isoform appears mutated varies by tissue type: a large percentage of adenocarcinoma of the lung (32%), pancreas (86%), and colon (41%) is driven by K-Ras mutations; 29% of melanomas are driven by mutations in N-Ras, while H-Ras mutations appears mutated in 5% of head and neck squamous cell carcinoma and 6% of bladder cancers [11].

## 2. Ras Activation and Downstream Signaling

Ras signaling is activated by cellular receptors including receptor tyrosine kinases (RTKs), G-protein coupled receptors (GPCRs), cytokines receptors, and extracellular matrix receptors [12]. Ras proteins act as molecular switches that cycle between two conformational states: an active GTP-bound state and an inactive GDP-bound state. This GDP/GTP cycling is tightly controlled by two main classes of regulatory proteins: guanine-nucleotide-exchange factors (GEFs), which upon receptor stimulation promote the activation of Ras proteins by stimulating GDP for GTP exchange, and GTPase-activating proteins (GAPs), which stimulate Ras-mediated GTP hydrolysis and Ras inactivation [6,12,13]. GTP binding to Ras induces changes in conformation, mainly in two regions named switch I and switch II, that greatly increase the affinity of Ras for its downstream effectors [14,15].

Ras-GTP stimulates a wide range of downstream effectors, although the best characterized are mitogen-activated protein kinases (MAPK), phosphoinositide-3 kinase (PI3K) [16,17], and the Ral pathways [18] (Figure 1). The Raf–MEK–ERK signaling axis was the first Ras-effector that was described [19,20] and it regulates cell growth, differentiation, inflammation, and apoptosis [21]. Activated Ras promotes the translocation of the Raf serine/threonine kinase to the plasma membrane, where it is activated and phosphorylated by different protein kinases. Active Raf phosphorylates and activates the MEK1/2 kinase, which in turn phosphorylates ERK1/2 mitogen-activated protein kinase that eventually exert their function on a large number of downstream molecules [22]. The phosphatidylinositol 3 kinase (PI3K) pathway is the second best-characterized Ras effector and participates in the regulation of a wide range of cellular activities, including cell growth, proliferation, differentiation, migration, and apoptosis [23]. Genetic alterations in PI3K/AKT and Raf/MAPK/ERK pathway components leading to aberrant activation of signaling are frequent in cancer [21,24]. The functional importance of the Ras–PI3K pathway and its role in cancer will be extensively discussed in the next sections.

Apart from the above-mentioned effectors, an increasing number of molecules that specifically interact with Ras have been described, such as Ral-GEF, Rho-GTPases, Novel Ras effector 1A (NORE1A), Af6, phospholipase C (PLC), Ras and Rab interactor 1 (RIN1), T-cell lymphoma invasion and metastasis-inducing protein (Tiam), and growth factor receptor 14 (Grb1), although the precise physiological role is still not fully understood for many of them [25]. Recently, Ras signaling has been shown to interact with the Hippo-signaling effector YAP to facilitate tumorigeneses [26]. Thus, the list of Ras effectors continues to grow, and more interactors belonging to the family of Ras effectors are expected to be characterized in the next years [14,27,28,29,30].

## 3. PI3K Family

PI3Ks are a family of lipid kinases with a key role in the regulation of cellular functions such as development, cell growth, metabolism, mobility, apoptosis, and proliferation [31]. Genetic deregulation of PI3K activity is associated with many human conditions such as allergy, inflammation, diabetes, neurological disorders, heart disease, or cancer [32,33].

PI3Ks are grouped into three different classes (Class I, II, and III) on the basis of protein structure, tissue expression, and substrate preference. These classes share four homologous regions, although the kinase domain is the most conserved [34]. Class I PI3K is the best-characterized family and the one most clearly implicated in human cancer [24], although much remains to be learned about their coupling to upstream signals and their relative functional output. Class I PI3K is composed of heterodimers formed by a catalytic subunit, which contains the kinase domain that catalyzes production of phosphatidylinositol (3,4,5)-trisphosphate (PIP3), and a regulatory subunit, which interacts with the catalytic subunit to downregulate substrate access and kinase activity [35]. Class I PI3K is further subdivided into Class IA and Class IB PI3Ks. The catalytic subunit of Class IA consists of p110α, p110β, or p110δ, which in humans are encoded by *Pik3ca*, *Pik3cb,* and *Pik3cd*, respectively. It can bind any of five different p85-like regulatory subunits (p85α, p55α, p50α, p85β, and p55γ). Class IB PI3Ks feature the p110γ catalytic subunit, which differs from the Class IA PI3K in its extreme N-terminus (lacking a p85 binding site) and binds either a p84 or p101 regulatory subunit. P110α and p110β isoforms are commonly expressed in all tissues, p110δ is mainly expressed in immune cells, and p110γ is expressed in immune cells and at low levels in the heart, pancreas, liver, and skeletal muscle [32,34,36].

Class I PI3Ks are activated through different upstream mechanisms, such as RTKs, GPCRs, or GTPases, which induce recruitment of PI3K to the membrane (Figure 2) [23]. Engagement of the ligand to its receptor induces the dimerization and autophosphorylation at tyrosine residues of the RTKs and subsequent interaction with Src homology 2 (SH2) domain–containing molecules [37,38]. From here, PI3K could be activated in three different ways: (i) by direct binding of the regulatory subunit of PI3K, p85, to phospho-YXXM motifs (X indicates any amino acid) within the RTK [39], triggering activation of the p110 catalytic subunit of PI3K; (ii) by Growth Factor Receptor-bound Protein 2 (GRB2) mediated activation, in which GRB2 binds preferentially to phospho-YXN motifs of the RTK [40] and to the scaffolding protein GAB, which in turn can bind to p85; (iii) via binding to Ras: GRB2 binds and activates Son of Sevenless (SOS), which then activates Ras, and in turn activates p110 independently of p85. GRB2 can also exist in a large complex that contains SOS, Ras, and GAB or other scaffolding proteins, bringing these activators into close proximity to p110 PI3K [41]. It remains unclear which of these pathways predominates in different physiological situations as data show that GPCRs, RTKs, and Ras proteins exhibit a wide level of plasticity in terms of activating the various Class I PI3Ks [23]. Only p110α, p110δ, and p110γ isoforms are activated by the Ras subfamily of proteins, while p110β is activated by members of the Rho GTPase subfamily, specifically by Cdc42 and Rac1 [17,23,42,43]. PI3K isoforms also show differences when activated by direct binding to RTKs or GPCRs. In the case of RTKs-dependent, Ras-independent activation, the Src Homology 2 (SH2) domain of the regulatory subunits of PI3Kα, PI3Kβ, and PI3Kδ, can bind to phosphotyrosyl residues of RTK or adaptor proteins such as insulin receptor substrate (IRS1) [23,44]. GPCRs only activate PI3Kγ and PI3Kβ isoforms as a result of direct interaction of the regulatory subunit and the Gβγ subunit of the heterotrimeric G proteins, following activation of GPCRs [23,35,45,46,47]. The different mechanisms by which each isoform is activated may have a significant effect on the pathological and physiological processes it mediates. A rigorous review of upstream regulators of PI3K and their role in disease has been published by Wang and colleagues [35].

Once activated, Class I PI3K phosphorylates the 3-OH group of phosphatidylinositol 4,5-bisphosphate (PIP2), resulting in the generation of PIP3 [48] that acts as a second messenger and recruits proteins that contain a pleckstrin homology (PH) domain, such as serine–threonine kinase AKT and its activating kinase 3-Phosphoinositide-dependent protein kinase 1 (PDK1), to the cellular membrane [49,50]. PDK1 phosphorylates AKT at T308, leading to partial AKT activation. Phosphorylation of AKT at S473 in the carboxy-terminal hydrophobic motif by the mammalian target of rapamycin complex 2 (mTORC2), fully activates AKT [22,51,52]. Phosphorylated AKT subsequently phosphorylates a large number of downstream targets that play an important role in the regulation of apoptosis and cell survival (Figure 2).

Phosphorylated AKT activates the mammalian target of rapamycin (mTOR) via phosphorylation and inactivation of proline-rich AKT substrate of 40 kDa (PRAS40) and tuberous sclerosis protein 2 (TSC2) [53]. mTOR phosphorylates 4E binding protein 1 (4EBP1) and ribosomal protein S6 kinase (S6K1), responsible for ribosomal S6 protein (S6/RPS6) phosphorylation [37] to promote protein synthesis and cellular proliferation [54]. AKT exerts an inhibitory phosphorylation on the glycogen synthase kinase-3 (GSK-3), inducing the expression of cell-cycle regulators such as c-Myc, cyclin D1, and cyclin E and thus promote cell cycle progression [55]. Another important target of AKT is the proapoptotic forkhead box transcription factors (FOXO) FOXO1, FOXO3, and FOXO4. AKT phosphorylates them, blocking the transcription of target genes that promote apoptosis (such as BIM), cell-cycle arrest (such as p21 and p27), and metabolic processes (such as sestrin3 and G6PC) [55,56]. AKT controls apoptosis through the inhibition of the proapoptotic activity of the BAD and BAX factors, from the the B-cell lymphoma 2 (BCL-2) family [57,58,59]. The murine double minute 2 (MDM2) is also phosphorylated and activated by AKT. MDM2 negatively regulates p53, which promotes cell survival [60,61]. Importantly, AKT activates the IKK/NF-κB signaling pathway, promoting many steps of cancer initiation and progression [55,62,63,64].

TEC family tyrosine kinases such as Bruton’s tyrosine kinase (BTK) and Targeting Interleukin-2-Inducible T-cell Kinase (ITK) are also PI3K effectors with a key role in lymphocytes [65]. BTK has a critical role in B-cell function and malignancy [66]. BTK is involved in the regulation of the developmental progression of pre-B-cells [67], and it is essential for B-cell receptor (BCR)-mediated proliferation and survival of mature B-cells [68,69]. Mutation of genes encoding BTK are implicated in human immunodeficiency disease X-linked agammaglobulinemia (XLA) [70,71], while BTK is abundantly expressed in B-cell leukemias and lymphomas [72]. The initial link between PI3K, BTK, and B-cells was provided by mouse knockout studies showing that the deletion of *Pik3r1* or *Pik3cd* caused defects in B-cell development and survival similar to those in mice lacking BTK [73]. Furthermore, pharmacological inhibitors of BTK and p110δ have shown a strong convergence of clinical activity in cancer, with best responses in malignancies of mature B-cells [74,75,76,77]. PI3K-dependent BTK signaling is a key example of a PI3K network that has emerged as an effective therapeutic target in cancer [78]. 

PI3K also targets GEFs involved in the activation of Rho/Rac/cdc42 GTPases family proteins [3,82]. For example, both pharmacological and genetic PI3K-α inhibition in endothelial cells reduced RhoA activation, which correlated with a migration and tail retraction defect [83]. In epithelial cells, growth factors stimulate the PI3K-dependent activation of Rac, leading to disruption of the actin cytoskeleton, release of filamentous actin-bound aldolase A, and an increase in aldolase activity [84].

PI3K pathway activity is negatively regulated by the phosphatase and tensin homolog (PTEN), a phosphatase that dephosphorylates and hydrolyses the secondary messenger PIP3, converting it back to PIP2. Loss of function mutations in PTEN result in enhanced PI3K signaling, which is associated with oncogenic cellular transformation and cancer [23,85]. Similar to PTEN, the SH2 domain-containing enzyme inositol 5′-phosphatase (SHIP) is also a negative regulator of PI3K, which specifically hydrolyses the 5-phosphate group from PIP3 [35]. A tumor suppressor role of SHIP1 has only been described in a single murine B-cell lymphoma model driven by oncogenic c-Myc and in lymphatic metastasis of breast cancer; no additional studies have demonstrated a tumor suppressor role of SHIP1 in other spontaneous malignancies in humans [86,87].

## 4. Ras–PI3K Interaction

Ras-mediated PI3K activation in response to growth factors requires two steps, the first of which is phosphorylation of the RTK and, in some cases, adaptor proteins, and secondly, activation of Ras small GTPases. Interaction between Ras and Class I PI3K is mediated by the binding of the Ras binding domain (RBD) present in the catalytic subunit of PI3K and the Ras switch region I (SW1) and II (SW2) that form part of the effector lobe of Ras [17,88]. Upon GTP loading of Ras, both switch regions suffer significant structural changes, increasing the affinity of Ras for effector targets [49,89]. Direct interaction between GTP-bound Ras and p110α augments the activity of p110α, possibly by inducing a conformational change at the substrate binding site, stimulating its catalytic activity [23] or by mediating a closer interaction with the plasma membrane [17,90]. Recent studies suggest that, at least for the RTK-induced activation of mammalian PI3K, Ras mostly acts to stabilize PI3K at the plasma membrane, whereas PI3K is allosterically activated by the RTK [91,92].

The multidomain architecture in PI3K regulatory and catalytic subunits contributes to the mechanisms through which PI3Kα activity is controlled as interactions between both subunits stabilize the overall structure and keep PI3K in the inactive state [93,94] (Figure 3). The N-terminal-SH2 (nSH2) and the Inter-SH2(iSH2) domains of p85 are required for full inhibition of p110α lipid kinase activity [95,96]. When the ligand binds to RTK, the phosphorylated tyrosine (pY) motifs in the receptors bind to the nSH2 domain of p85, thus disrupting the inhibitory contact between the nSH2 and the catalytic subunit and inducing a conformational change in p110α that leads to its activation [95]. Specifically, after nSH2 release, the C-lobe of the kinase domain moves away from the C2 domain, exposing the p110α membrane binding surface to membrane interactions. The activation loop in the kinase domain becomes more flexible and approaches ATP, generating an active PI3Kα conformation for substrate catalysis [36] (Figure 3). For Ras-mediated PI3K activation, the inhibition exerted by p85 is relieved through dissociation of p110 and p85, with the nSH2 domain of p85 playing a key role [96,97] (Figure 3). Recently, computational analysis of K-Ras4B-mediated activation of p110α has shown that K-Ras4B/RBD binding disrupted the interactions along the p110/p85 interface through the induction of overall structural displacement of both nSH2 and iSH2 in p85 from the neighboring functional domains in p110 including the C2, helical, and kinase domains. This results in the exposure of the kinase domain and facilitates the full activation of PI3Kα [98].

Ras-mediated regulation of PI3K plays a role in many cellular processes involved in both normal physiology and disease [61]. In all these contexts, Ras-stimulated production of PIP3 results in membrane recruitment and, in many cases, activation of PH domain-containing proteins, including PDK1 and AKT kinases, GEFs for Rac small GTPases, and BTK. The best-characterized roles of Ras–PI3K signaling in eukaryotes are the regulation of AKT function in cell survival and growth, as well as the remodeling of the actin cytoskeleton [99,100]. The Ras–PI3K pathway can also regulate the actin cytoskeleton by promoting Ras signaling in various cellular contexts [91,101].

### 4.1. Ras–PI3K Interaction in Cell Growth and Apoptosis

Significant efforts have been made to understand the role of Ras and PI3K in the regulation of different cellular events and its involvement in cancer since the discovery of their interaction [102]. PI3K activation by different upstream receptors makes it difficult to uncover the specific role of Ras-dependent activation of PI3K in the control of physio-pathological processes. Adding to this complexity, Ras can activate a large number of effectors, each of them contributing to the regulation of a specific cellular event. Thus, fully understanding of the specific roles played by this interaction has been challenging.

Different approaches have been used to examine the functional relevance of Ras-dependent activation of PI3K. Traditionally, the Ras–PI3K signaling pathway was studied using genetic animal models or cell lines in which Ras was constitutively activated or silenced [103,104,105,106], and AKT phosphorylation was used as a readout for Ras-dependent PI3K activation. These tools were usually complemented with pan- or isoform-specific PI3K and ERK small molecule inhibitors to allow the evaluation of the individual roles of Ras effectors in the final physiological response [107,108]. The general conclusion drawn from these studies was that Ras requires PI3K activation to regulate cell viability, through the control of proliferation and apoptosis [65,104,109], and to induce transformation of epithelial cells [99,103]. Thus, it was concluded that dysregulation of the processes controlled by the Ras–PI3K pathway or mutation on any of its components leads to oncogenesis.

The use of PI3K inhibitors to study Ras-downstream signaling has also been used to determine the functionality of this effector pathway. However, these inhibitors block PI3K activity independently of how it is activated. To circumvent this issue, Rodriguez-Viciana et al. designed different Ras mutants that are specifically unable to interact with each of its effectors [100]. This study showed that mutations in different residues of the effector-binding domain of Ras altered the transformation capability of oncogenic Ras and that both Raf and PI3K activation are critical to induce fibroblast transformation by Ras. Additionally, it demonstrated that PI3K, but not Raf, modulates membrane ruffling and cytoskeletal reorganization. Subsequent studies using these mutants showed that Ras–PI3K interaction protected fibroblasts from c-Myc-induced apoptosis as effectively as constitutive activation of p110α, and this protection was abolished by the treatment of fibroblasts with a PI3K inhibitor [109].

Important advances in the understanding of the specific role of Ras-dependent activation of PI3K were achieved by the generation of different mouse models in which Ras binding to the catalytic unit of PI3K was prevented. Suire et al. generated a mouse model in which five residues of the Ras-binding domain of p110γ were altered (T232D, K251A, K254S, K255A, and K256A). Using this model, they established that Ras interaction with p110γ is not necessary for mouse development, since mutant mice were viable, fertile, of normal size, and with generally normal blood counts. However, the study showed that neutrophils in which p110γ cannot interact with Ras showed reduced PIP3 accumulation, AKT phosphorylation, chemokinetic responses, and reactive oxygen species (ROS) production in response to GPCR agonists [101].

Studies in *Drosophila* showed that Ras–PI3K interaction is essential during development [110,111]. Substitution of four amino acids in the Ras binding domain of Dp110 (the only Class I PI3K in *Drosophila*) (T231D, K250A, R253A, and K257A) prevented its interaction with Ras. The resulting mutant flies were viable, but smaller in size. Female flies were less fertile than wild-types (WTs), pointing to a role of Ras/PI3K interaction in the response to growth-stimulating stimulus. Furthermore, AKT activation was diminished in response to insulin in the brain and imaginal discs, and basal AKT activation was downregulated in the ovaries [110]. Further studies using this model showed that Ras activation of PI3K is also required to decrease motor neuron excitability, but not for the PI3K-dependent increase in nerve terminal growth [112].

Generation of a mouse model in which the binding of Ras to the p110α subunit of PI3K was prevented by the introduction of two point mutations (T208D and K227A) in the RBD domain of the endogenous *Pik3ca* gen revealed additional roles of Ras-dependent activation of PI3K. In these mice, RBD mutations did not affect the basic enzymatic activity of p110α, coupling to the p85 regulatory subunit, expression of PI3K components (p85, p110α and p110β), or interaction with other Ras effectors. However, the homozygous mutant mice died shortly after birth due to deficient lymphatic vasculature development, and the few surviving mutant adults exhibited reduced body weight compared to WT [113]. The importance of a functional p110α subunit in vascular development during embryogenesis was also shown with a mouse model where ubiquitous or endothelial cell-specific inactivation of p110α led to embryonic lethality at mid-gestation, associated with severe defects in the vascular remodeling [83]. Conversely, mice with inactive p110β or p110γ were viable and fertile, without obvious vascular defects, confirming that p110α plays a unique role among Class IA PI3K isoforms in embryonic vascular morphogenesis and remodeling [83] and that this effect may be mediated by Ras. These results also suggested that cellular responses during embryogenesis and adulthood require different growth signals, and Ras activation of PI3K may have a function in the regulation of cell responses to different growth stimulus. This idea is reinforced by the observation that, in fibroblasts lacking Ras–PI3K interaction, epidermal growth factor (EGF) and fibroblast growth factor 2 (FGF2) fail to properly induce AKT activation, whereas platelet-derived growth factor (PDGF) does not [113,114], and migration is impaired in response to EGF, FGF2, hepatocyte growth factor (HGF), and insulin but not in response to PDGF [115]. Furthermore, insulin-induced activation of AKT and ERK is also reduced in hepatocytes in which *H-ras* is silenced [116], indicating that different Ras isoforms may also differentially regulate growth factor-induced signaling pathways.

It is well established that PI3K is one of the principal effectors of Ras in the regulation of the cell cycle, being the main effector promoting transition from G0 to G1 phase in quiescent cells [117,118,119], as well as entrance of cells into S phase [117]. In parallel, Ras regulates pro-survival signals via PI3K activation of AKT. Phosphorylated AKT, in turn, phosphorylates a number of substrates that play a role in the regulation of apoptosis, such as BAD [57,120,121], NF-kB [122,123,124], or FOXO [125]. Thus, by stimulating proliferation and preventing apoptosis, Ras–PI3K signaling supports tumor growth.

The role of PI3K signaling in apoptosis inhibition is not restricted to the classical Ras isoforms. R-Ras is a Ras family member that functionally differs from H-Ras, N-Ras, and K-Ras [126] and is highly expressed in the vascular endothelial cells [127], where it promotes lumenogenesis, supports the lumen structure with the stabilized microtubules [128], and promotes vessel maturation by endothelial barrier stabilization and pericyte association [129,130]. Using human umbilical vein endothelial cells (HUVEC) cells, Takino et al. described that Ras guanyl nucleotide releasing protein 2 (Ras-GRP2), a Ras GEF, inhibits apoptosis by inhibiting BAX activation in a PI3K-dependent way [131]. The authors demonstrated that Ras-GRP2 suppresses BAX activation-induced apoptosis by promoting HK-2 translocation to mitochondria via R-Ras/PI3K/AKT signaling pathway. The BAX pathway is involved in apoptosis in endothelial cells in conditions of hyperglycemia and high methylglyoxal levels as a trigger of atherosclerosis and in lipopolysaccharide-induced apoptosis caused by inflammation [132,133]. Therefore, inhibition of BAX translocation by Ras-GRP2/R-Ras/PI3K/AKT/HK-2 may result in a survival benefit in these conditions.

### 4.2. Ras–PI3K Signaling in the Immune System

Most studies on the role of PI3K in the immune system focus on PI3Kγ and PI3Kδ isoforms, while the function of PI3Kα in immune cells is less understood. The importance of PI3Kγ and PI3Kδ isoforms for the normal activity of the immune system has been revealed in mice lacking functional versions of PI3Kγ [47,134] or PI3Kδ [135,136], which are viable but show significant defects in both innate and adaptive immunity [137]. Mice lacking 110δ exhibit impaired B-cell development and humoral immune responses, reduced T-independent antibody responses, and relatively normal numbers of thymocytes [135,136,138], while p110γ-deficient mice have a modest reduction in thymic cellularity and no alterations in development or function of B-cells [134].

The contribution of Ras signaling to 110δ or p110γ in the maturation of the immune system during development has not been determined; however, results from studies using N-Ras knockout (KO) mice overlap with those obtained with p110γ deficient mice [134,139]: both N-Ras and p110γ regulate proliferation and IL-2 production of T-cells and promote an inflammatory response after stimulation with an infectious agent.

PI3Kγ can operate as a p110γ/p84 or p110γ/p101 complex, which confers PI3Kγ signal-specificity [140]. These complexes show some differences in tissue distribution. The p84 subunit is highly expressed in the immune system but also in other tissues and organs, such as the heart, pancreas, and the central nervous system, while p101 shows a more restricted expression profile in the immune system but was below the detection limit in other body compartments [141,142,143]. Such differences of expression and distribution in human tissues point to different cellular functions of both enzymes. In fact, p84 and p101 regulatory subunits show a distinct function in specific cell types, such as mast cells, neutrophils, and cardiomyocytes [142,143,144,145]. For example, neutrophils lacking p84 have selective defects in p110γ-dependent oxidase activation, but neutrophils lacking p101 show selective defects in p110γ-dependent motility [145]. Given the different role that p84 and p101 can play in the control of cellular functions, it seems indispensable to understand the mechanisms that drive their activation.

The PI3Kγ isoform can be activated by GPCRs or by Ras-dependent mechanisms [35], but it is unclear which of the two mechanisms is more relevant [146]. There is a general agreement that p101/p110γ is more sensitive to activation by Gβγ than p84/p110γ, which may be due to a decrease in the affinity of p84 to Gβγ [142,147]. In agreement with this, studies in HEK cells have suggested that Ras is an indispensable co-regulator of p110γ/p84, which requires Ras co-stimulation for full activation, whereas p110γ/p101 activation is mainly mediated by interaction with the Gβγ subunit of GPCRs, without any input from Ras [140]. In apparent contradiction to this, data obtained with primary neutrophils bearing mutations that prevent interaction of Ras with the RBD domain of p110γ show that Ras-GTP is an important regulator of p101/p110γ and not only of p84/p110γ [101,145]. These data illustrate that p84 and p101 variants of PI3Kγ are stimulated by Ras and Gβγ to different degrees depending on cell type, stimulus, and Ras isoform involved and that these differences shape cellular responses.

Ras–PI3K signaling has been shown to regulate different functions in neutrophils. Neutrophils in which PI3Kγ cannot bind to Ras (p110γ^DASAA/DASAA^) and p101^−/−^ neutrophils (used as a specific mean of assessing the role of Gβγs in the activation of PI3Kγ) exhibited a reduction of PIP3 accumulation, AKT activation in response to GPCR agonists, and reduced migration. However, loss of Ras binding to PI3Kγ specifically reduced neutrophils’ ability to produce ROS after stimulation with the peptide N-formyl methionyl–leucyl–phenylalanine (fMLP) or the complement fragment (C)5a (C5a) [101]. Further investigation confirmed that, at least in neutrophils, Ras induces ROS formation via two Ras-GEFs: Ras guanyl releasing protein 4 (Ras-GRP4) was required for fMLP-stimulated ROS formation in unprimed neutrophils, while SOS1/2 have a role in primed neutrophils [148]. In vivo studies demonstrated that Ras signaling in the absence of GPCR stimulation is not sufficient to activate PI3Kγ, and Ras interaction with p110γ is required for full GPCR stimulation of PI3Kγ, suggesting that Ras combines synergistically with inputs from Gβγ to determine PI3Kγ activity [149].

Ras signaling to PI3K is also relevant for the control of allergies. Mast cells are key effectors in the pathology of allergic disease [150]. In mast cells, the pharmacological inhibition of Ras with a farnesyltransferase inhibitor (FTI-227) reduced AKT phosphorylation upon activation with adenosine. Functionally, FTI-277 attenuated degranulation of IgE/antigen-activated mast cells upon co-stimulation with adenosine, reduced expression of tumor necrosis factor α (TNF-α) and IL-6, and impaired cell migration [146]. These data suggest that the development of specific p110γ/p84 targeting strategies for mast cell-related diseases will presumably have limited effects on p101-dominated immune cells, thus reducing side effects.

## 5. Ras–PI3K Interaction in Cancer

Mutations in the Ras genes were first reported in cancer over 30 years ago and numerous studies have since validated mutant Ras as a driver of tumor initiation and maintenance [151]. Activating point mutations in the genes encoding the Ras subfamily are found in approximately 27% of all human cancers, with 98% of the mutations occurring at one of three mutational hotspots (G12, G13, and Q61), regardless of the isoform involved [5,10]. These mutations disrupt GAP-mediated GTP hydrolysis, which results in an accumulation of constitutively GTP-bound Ras in cells and an exacerbated activation of effector pathways even in the absence of upstream stimulation [10,11]. Additional mechanisms of Ras activation in cancer include (i) perturbation in GDP–GTP regulation; (ii) loss of GAPs, such as neurofibromin (NF1), and (iii) persistent RTK-mediated activation of GEFs [5].

PI3K also plays a critical role in cancer. Mechanisms that enhance PI3K activity include mutations in proteins upstream of PI3K (such as Ras or RTKs), mutational activation of PI3K itself (mainly mutations in the *Pik3ca* gen), or loss of function of its inhibitor PTEN [23]. In particular, mutations in *Pik3ca* are the second most common alteration in human tumors [35,152]. Gene insertions, deletions, and somatic missense mutations in *Pik3ca* are frequent in colon, breast, brain, liver, stomach, and lung cancers [153]. There are two common “hotspots” for mutations in *Pik3ca*, H1047R, which enhances the interaction of the kinase domain with membranes and bypasses the requirement for association with Ras [154], and E525/E545K, which disrupts the inhibitory interface with the n-SH2 domain of the regulatory subunit [23,95,155]. Other mutations that activate PI3K occur in the *Pik3r1* and *Pik3r2* genes, coding for the regulatory subunits p85α and p85β, respectively. Mutations in these genes reduced the ability to interact with and stabilize the p110 catalytic subunit in type IA PI3Ks [156]. Aberrant PI3K activation can also occur due to loss-of-function mutations and deletions in PTEN, which is the third most frequent alteration in human tumors [35]. Ras and *Pik3ca* mutations are mutually exclusive in endometrial and breast cancers, but co-exist in colorectal and lung tumors [31,157,158,159], suggesting that the acquisition of both mutations may confer additional characteristics that are beneficial for the tumor. For example, in colorectal cancer, coexistant K-Ras and PI3K mutations decrease sensitivity to PI3K and mTOR inhibition [158], and in lung cancer, the addition of a constitutively active PI3K mutant to epidermal growth factor receptor (EGFR) mutant tumors confers gefitinib resistance in vitro [160]. Thus, determining *Pik3ca* mutation status and its coexistence with other gene mutation may be helpful to predict response to targeted therapy.

The initial indications that Ras interaction with PI3K could regulate tumorigenesis were obtained by in vitro studies showing its role in oncogenic cell transformation and cell survival [65,100,103,161]. Since then, a large number of studies have shown that PI3K activation regulates different aspects of Ras transformation [162,163]. To provide some specific examples, one study showed that activation of PI3K/AKT pathway is involved in intestinal epithelial cell transformation following induction of oncogenic Ha-RasVal12: pharmacological inhibition of PI3K activity exerted an anti-neoplastic effect on Ras-expressing intestinal epithelial cells, which included stimulation of apoptosis, reversal of morphological transformation, and accumulation of cells in the G1 phase of the cell cycle [103]. PI3K is also required for Ras transformation of fibroblasts and acts upstream of Rac GTPase to induce cortical actin rearrangement, membrane ruffling, and lamellipodia formation in oncogenic Ras-transformed cells [100]. Using a dominant negative mutant of Ras and PI3K inhibition, it was demonstrated that Ras and PI3K activation participates in ghrelin-caused proliferation of human colon cancer cells through the activation of the AKT/mTOR pathway [65]. Moreover, Ras–PI3K signaling regulates prostaglandin E2 inhibition of apoptosis in cancer cells [161].

In vivo, Ras activation of PI3K signaling is essential for lung carcinogenesis driven by oncogenic mutation of K-Ras and the formation of skin tumors induced by activating mutations of H-Ras [113]. However, once tumors are established, they are less dependent on RAS-PI3K signaling and disruption of their interaction only leads to partial tumor regression, followed by long term stasis [164]. Similar results were obtained in lung tumors developed in response to K-Ras activation with normal PI3K signaling, in which established tumors did not reduce in size when mice were treated with NVP-BEZ235, a dual pan-PI3K/mTOR inhibitor, even though tumors driven by expression of activated PI3K did regress upon treatment with this drug [160]. These data suggest that additional K-Ras activated pathways cooperate with Ras to promote tumor maintenance and development. Combination of PI3K and MEK inhibition caused impressive tumor regression, albeit accompanied by significant toxicity [164,165]. Similarly, subcutaneous tumors formed by tumor cells in which Ras preferentially binds to PI3K continue to grow even in the absence of oncogenic Ras signaling. However, if tumors are formed by cells in which Ras preferentially binds to Raf or RalGEF, they stop growing or even regress in some instances. In this setting, the activation of the three pathways was required for Ras-induced tumor initiation, suggesting that Ras-dependency of downstream pathway activation may vary with tumor stage and tumor type [166]. Ras–PI3K interaction is also required for tumor progression of WT Ras cancers since removal of Ras–PI3K interaction in an EGFR mutant lung cancer model caused important tumor regression [114]. These results suggest that direct targeting of the Ras-binding domain of p110α may be effective in K-Ras-driven lung tumors and in WT-mutant lung tumors. According to the observation that PI3K mediates oncogenic signaling in Ras-WT cancers, Molina-Arcas and colleagues showed that PI3K is critical for survival of both K-Ras mutant and K-Ras WT NSCLC cells, as PI3K inhibition induced loss of cell viability irrespective of the genotype [167]. However, opposite results on the role of oncogenic Ras on PI3K activation were obtained in colorectal cancer cells carrying mutations in K-Ras, where knockdown of K-Ras did not suppress AKT phosphorylation and PI3K/AKT pathway required insulin-like growth factor I receptor (IGF-IR)-induced activation. This is therapeutically important as, while suppression of mutant K-Ras is not sufficient to downregulate PI3K/AKT, IGF-IR inhibition could suppress PI3K signaling in K-Ras mutated colorectal cancers [168]. In summary, PI3K is not under the sole control of mutant K-Ras, and consequently, to discern the mechanism of PI3K activation in different cancers seems to be essential for the development of new drugs.

p110α is also critical for K-Ras-driven pancreas carcinogenesis as p110α inactivation prevents mouse lethality and the appearance of all types of pancreatic lesions induced by mutated K-Ras [169]. Supporting these data, Eser et al. demonstrated that oncogenic *Pi3kca* mouse models show similar patterns of acinar-to-ductal metaplasia (ADM), pancreatic cancer progression, and activation of key downstream effectors of PI3K to the K-Ras G12D model, indicating that PI3K signals downstream of mutated K-Ras in pancreatic cancer [170]. Further, PDK1, AKT, or PI3K inhibition resulted in normal life expectancy and inhibition of pancreas cancer development in the K-Ras G12D model.

The array of proliferative signals generated by oncogenic Ras culminates with the upregulation of several transcription factors that are required for cell cycle entry and progression, including FOS, serum response factor (SRF), the leucine zipper protein JUN, the ETS domain-containing transcription factor ELK1, activating transcription factor 2 (ATF2) and NF-κB [171,172,173,174,175,176]. In turn, these factors trigger the expression of cyclin D1 [177,178,179]. Although initial studies attributed the stimulation of cyclin D1 transcription by Ras to the activation of the Raf–MAPK pathway, it has become evident that additional levels of control are achieved through the Ras activation of PI3Ks, the RHO family of GTPases Rac1, and the Ral-guanine nucleotide dissociation stimulator (GDS) family of GEFs [19,177,178,180,181,182,183]. In addition to stimulating cyclin D1 gene transcription, oncogenic H-Ras regulates the metabolic stability of the cyclin D1 protein through PI3K-dependent inhibition of GSK3β, the kinase that is responsible for the phosphorylation and the consequent ubiquitylation and proteasomal degradation of cyclin D1 [184].

Numerous molecular abnormalities that result in constitutive activation of the PI3K pathway have been reported in hematologic malignancies, demonstrating the importance of targeting PI3K in leukemia and lymphomas [185]. However, there are few data describing the requirement of PI3K activation in Ras-mutated hematological malignancies. A possible link between Ras-p110γ interaction and the development of T-cell acute lymphoblastic leukemia (T-ALL) was shown by Janas et al. [186]. The authors demonstrated that thymocytes harboring a mutation of p110γ that blocked its interaction with Ras have a reduced proliferation rate in response to CXC Motif Chemokine Receptor 4 (CXCR4, a receptor implicated in β-selection of thymocytes) and consequently were defective when undergoing β-selection in comparison with WT counterparts. Thus, in thymocytes, p110γ is a key target of Ras signaling that is required for an optimal proliferative response at β-selection, and therefore this pathway may be implicated in the development of T-ALL, which typically presents the expansion and dissemination of cells arrested between β-selection and double positive stages. p110α may also have a role in Ras-driven leukemia, since the loss of p110α caused a significant improvement in survival in a murine model of leukemia induced by oncogenic K-Ras mutations and prolonged the latency of the myeloproliferative neoplasm [105].

As metabolic disorder was pointed out as a hallmark of cancer, great attention was paid to the crucial role of oncogenic Ras in tumorigenesis, where it orchestrates a metabolic reprogramming of tumors [187]. Oncogenic Ras plays an important role in regulating cancer cell metabolism by triggering several main metabolic changes [188]. Activation of Ras upregulates growth-promoting pathways controlled by PI3K-mTOR, and growing evidence suggests that there is a tight correlation between metabolic rewiring in cancer and these pathways. For example, Ras mutant cells are dependent on sufficient glucose uptake and PI3K can increase glycolysis through activation of AKT and stabilization of Hypoxia-inducible factor 1 (HIF-1) [189,190]. Additionally, one of the effectors of PI3K, mTORC1, can drive a complex pathway to support glycolysis, the pentose phosphate pathway (PPP), and nucleotide biosynthesis [191,192]. Analysis of different genes with driver mutations that were linked to the regulation of metabolic pathways in K-Ras-driven cancers, such as *Pik3ca* [49], *KMT2D* [193], *PTEN* [194], and *IDH1* [195], suggested a theoretical broader transcriptional and signaling circuit coordinated by Ras together with *p110α*, NF1, and PTEN for the tight regulation of metabolic pathways [188]. Given the relevance of Ras–PI3K signaling in metabolic rewiring, the different levels of interplay between the PI3K–AKT and K-Ras–MAPK signaling cascades [164] should be considered, including for potential therapies. Additional studies are needed to understand how the cross-talk between these two pathways contribute to the metabolic rewiring needed to sustain uncontrolled proliferation in cancer [196].

### 5.1. Ras–PI3K Signaling in Migration and Metastasis

The metastatic spread of the primary tumor accounts for over 90% of cancer deaths [197]. Metastasis formation requires complex interactions between cancer cells and multiple microenvironments [198]. The epithelial-to-mesenchymal transition (EMT) process is a crucial mechanism in the progression of tumor metastasis and is characterized by the loss of cell–cell adhesions and the acquisition of cell–matrix interactions [199], which allow cancer cells to become mobile and invasive, enabling metastasis and chemotherapy resistance [197]. Both Ras and PI3K/AKT pathways are well documented to drive EMT, invasion, and metastasis [200,201,202,203,204,205]. However, the importance of Ras–PI3K interaction in Ras-induced EMT is not clear, and the signaling pathways by which Ras contributes to EMT are controversial [206]. On one hand, K-Ras silencing inhibited EMT and proliferation of breast cancer cells, while promoting apoptosis, but these effects were reversed when mutant cells were treated with a PI3K/AKT agonist [99]. Furthermore, PI3K-AKT signaling was suggested to be required for transforming growth factor β (TGF-β) induced EMT and cell migration in mammary epithelial cells [207]. In the same line, another study demonstrated that H-Ras-driven tumors induce PI3K/AKT-dependent ß-catenin activation and that this is associated with increased cell proliferation, EMT, and cell invasion [208]. These data would suggest that PI3K acts downstream of Ras to regulate EMT, proliferation, and apoptosis. Additional supporting evidence of the role of PI3K as a mediator of Ras-induced EMT was obtained in esophageal cancer cells, where depletion of Ras-GAP SH3 domain binding protein (G3BP1, a protein that modulate the Ras signaling pathway via interacting with the SH3 domain of RasGAP and found overexpressed in some cancers [209,210,211]) represses invasion and migration potential via EMT suppression. This was accompanied by a significant decrease in the levels of p-AKT and p-GSK3β. The overexpression of G3BP1 had the opposite effect, which was revered by pharmacological inhibition of PI3K [212]. On the other hand, other studies have established that PI3K is not required for Ras-induced EMT. Janda and colleagues demonstrated that while the PI3K pathway is required to induce rapid tumor growth and enhanced proliferation of mammary epithelial cells expressing oncogenic H-Ras in collagen gels, it fails to cause EMT in vitro and in vivo and EMT induction is mediated by MAPK-pathway activation [213,214]. According to these data, Fischer and colleagues showed that ERK/MAPK activity is necessary and sufficient for the cooperation with TGF-β to regulate proliferation and survival of hepatocytes during EMT and to promote hepatocellular tumor progression [206]. All together, these data suggest that the contribution of Ras effectors in the control of EMT may depend on the cell type, the cellular context, and the Ras isoform involved.

Acquisition of a migratory phenotype and extensive reorganization of the actin cytoskeleton is one of the first requirements for metastasis formation. Oncogenic activation of RAS has been implicated in facilitating almost all aspects of the malignant phenotype, and metastasis is not an exception [215,216]. Oncogenic Ras induces alterations in cell–cell and cell–matrix interactions and the acquisition of a migratory phenotype. The perturbation of cell–cell contacts by oncogenic RAS is accomplished through the targeting of the molecular machinery that maintains intercellular adhesion junctions, including the E-cadherin receptor and its associated cytoplasmic protein β-catenin [217,218]. Additionally, oncogenic RAS directly contributes to the enhanced motility of cancer cells by causing pronounced changes in the polymerization, organization, and contraction of actin, the polymerization and/or stability of microtubules, and the transcriptional regulation of mitogenic gene products [215,219]. Collectively, these changes establish the front–rear asymmetry that is required for cell migration.

The existence of several functional links between the Ras, PI3K, and Rho-mediated signaling pathways must be considered when assessing oncogenic Ras in metastasis [82,220,221,222]. Ras regulates the actin cytoskeleton through Rac, a process that is entirely dependent on normal PI3K function acting upstream of Rac [100]. Rac and PI3K are linked by a feedback loop that is critically involved in the establishment and maintenance of cell polarity. Ras-GTP activates PI3K and the production of PIP3 at the leading edge of migrating cells. This increase in PIP3 induces Rac activation [223], which in turn sustains the formation of directional protrusions. Active Rac binds to p85 and further stimulates the activation of PI3K, thus amplifying Rac activation and migration induction [224]. PI3K also activates Rac through interaction with the Rac-specific GEF Tiam1 [225], and PTEN^−/−^ cells are more motile and contain higher levels of Rac-GTP and Cdc42-GTP than WT cells [226]. In some studies, it was observed that even though PI3K is the main mediator, different Ras isoforms activate different Rho family members. For example, in Caco2-cells, oncogenic K-Ras enhances cell migration and filopodia formation through Cdc42, and oncogenic H-Ras induces EMT characteristics and promotes migration and invasion through Rac1 [227].

Inhibition of PI3K activity halts Ras induction of membrane ruffling, while activated PI3K is sufficient to induce membrane ruffling, acting through Rac [228]. Disruption of Ras binding to PI3K impairs migration in MEFs in response to EGF, FGF2, insulin, and oncogenic Ras, but not in response to PDGF, which has a diminished ability to activate Rac [115]. Furthermore, in response to EGF, cells deficient in Ras–PI3K interaction migrated in a disorderedly manner, suggesting a defect in polarization, and showed reduced invasion ability. The Ras–Raf scaffold protein Sur8 plays an important role in mediating motility and invasive potential of cells predominantly through the PI3K pathway via activation of Rac and MMPs, with a minor contribution of the Erk pathway [229]. Although PI3K activation is mainly linked to Rac activation, it has been reported that PI3K can also activate RhoA [83,230]. For example, endothelial cells with reduced PI3Kα activity have defects in migration, correlating with reduced levels of active RhoA [83].

Ras–PI3K interaction also regulates non-cell-autonomous cancer cell motility. Co-culture of Ras-transformed epithelial cells and normal epithelial cells showed that the majority of Ras-transformed cells are extruded towards the basal side of the surrounding normal cells, which is concomitant with enhanced motility [231]. The basally extruded Ras cells exhibited motility when surrounded by normal cells, but were non-motile when cultured alone, indicating that a non-cell-autonomous mechanism is at play. Inhibition of PI3K suppresses basal extrusion of oncogenic Ras cells, suggesting that the PI3K/AKT pathway is involved in the extrusion of oncogenic Ras cells to the basal side of normal epithelial cells. This data indicates that the role of Ras–PI3K in the regulation of metastasis is more complex than initially thought and therefore a complete characterization of the paracrine effects of K-Ras-mutant cancer cells would be valuable to fully understand its role in this process and to identify tumor specificities.

### 5.2. Ras–PI3K Signaling in Therapy Resistance

Drug resistance is the main limiting factor for achieving cures in cancer patients [232]. Activation of the AKT pathway is one of the major mechanisms involved in intrinsic and acquired resistance to radiotherapy and chemotherapy agents [233,234] (Figure 2). Due to the central role of AKT in cell survival, protein synthesis, and proliferation, an increase in AKT activity can evade the cytotoxic effect of chemotherapeutic agents, leading to chemoresistance [79]. This may be due to protecting cells from drug-induced apoptosis through the intrinsic pathway (e.g., by inactivating BAD and caspase 9 and stimulating anti-apoptotic proteins MCI-1) or through upregulation of survivin as an inductor of apoptosis [235,236].

Radiotherapy induces phosphorylation of AKT through PI3K activation [237,238,239], and the level of phosphorylation is radiation-dose dependent [240]. Double strain breaks (DSBs) are the major cause of IR-induced cell death in radiotherapy, and clinical data exists indicating the prognostic value of AKT activation levels for radiotherapy response in some tumor types [241,242]. Activated AKT stimulates the repair of radiation-induced DSBs [80,81,238,243] and the C-terminal domain of AKT interacts with the DNA-dependent protein kinase catalytic subunit (DNA-PKcs), the major component in non-homologous end-joining repair of DSB, and phosphorylates it, stimulating function of DNA-PKcs in DSB repair. Hyperactivation of AKT due to deregulation of RTKs, PI3K, Ras, or PTEN leads to efficient DSB repair and the appearance of radiotherapy resistance [244,245,246,247,248].

Although the specific contribution of Ras in the activation of PI3K/AKT leading to therapy resistance is not always addressed, in some cases it is known to play a relevant role [249]. For example, in lung carcinoma cell lines A549 and H460, targeting EGFR, PI3K [250], and AKT [244] enhances repair of DNA-DSBs and induces DNA-PKcs–dependent radiosensitization. These reports are supported by data from Choi et al., who showed that in the same cell lines, silencing the expression of K-RasK-Ras induces radiosensitization due to a reduction of IR-induced phosphorylation of DNA-PKcs and impaired repair of DNA-DSBs in a PI3K-AKT–dependent manner [251] due to enhanced autocrine production of EGFR ligands and activation of the Ras/PI3K axis [247]. This effect was not observed in K-RasK-Ras wild type cells.

Platinum-based agents, such as cisplatin, carboplatin, and oxaliplatin, are used in the treatment of a variety of cancers [79,252,253]. They are DNA-intercalating agents that interfere with DNA replication and RNA transcription through the crosslinking of DNA. Cisplatin also induces mitochondrial ROS, further increasing DNA damage and the cytotoxic properties of the drug. This results in the formation of DNA adducts, driving tumor cells to apoptosis [254,255]. There are many resistance mechanisms associated with cisplatin, including the involvement of oncogenicK-RasK-Ras mutations [256,257,258,259] and the hyperactivation of the PI3K-AKT pathway. K-RasK-Ras mutations induce *NRF2* transcription and pathway upregulation, resulting in the overactivation of the anti-oxidative stress pathway, rendering tumor cells resistant to cisplatin-induced ROS [257,260]. In lung cancer cell lines, SHP2 mediates cisplatin-resistance-related phosphatase in a process that involves the inhibition of apoptosis by the activation of Ras/PI3K/AKT/survivin signaling pathway, and, consequently, the inhibition of SHP2 was associated with reduced expression of Ras, AKT, AKT activation, and survivin [261]. In ovarian cancer cell lines, crosstalk between STAT3 and Ras/p53 activated PI3K and ERK pathways, which in turn inhibited endoplasmic reticulum stress (ERS)-associated molecules, blocking cellular autophagy and inducing cisplatin resistance [262].

ERK activation is a common feature of tumors with a K-Ras, N-Ras, or B-Raf mutation [263], and the inhibition of the Raf/MEK/ERK pathway was supposed to be effective in cancers harboring mutations in these genes [264]. However, a portion of patients developed drug resistance mechanisms and no longer responded to Raf or MEK inhibitors [265]. Upregulation of PI3K pathway was found to be a major mechanism of resistance to Raf and MEK inhibitors [266,267,268]. The activation of PI3K after ERK pathway inhibition comes from different mechanism that include RTK reactivation [268,269], activating mutations in *Pi3kca* or loss of PTEN [270], or activating a positive feedback loop composed of GAB1, Ras, and PI3K, which induces the bypass of the ERK signal to the PI3K signal [271].

Recently, small molecules against K-Ras G12C mutations have been developed, and the first drug targeting this mutation has been approved by FDA [272]. However, therapeutic resistance to K-Ras G12C inhibition has been observed in preclinical tumor models and also in the clinic [273,274,275]. The PI3K pathway may be implicated in the resistance to K-Ras G12C inhibitors [276] as preclinical studies have shown failure to inactivate the PI3K signaling pathway after treatment with G12C inhibitors [274,277,278]. Importantly, combination of G12C and PI3K pathway inhibitors was effective in vitro and in vivo on models that are resistant to single-agent G12C inhibitor [274], or significantly improved antitumor activity of G12C inhibitors [277,279], which could be explained by a concomitant inhibition of p-ERK (due to G12C inhibition) and p-AKT [274,277]. This combination could avoid the toxicity associated with the inhibition of MEK and PI3K, while the efficacy of inhibiting both pathways in tumor cells is maintained [279].

Tyrosine kinase inhibitors (TKIs) are used in a variety of cancers. Upregulation of Ras-mediated pathways is a common mechanism of resistance to TKIs [280,281,282,283,284,285], such as those targeting RTKs including FMS-like tyrosine kinase 3 (FLT3) and EGFR. Jacobsen et al. [286] showed that AKT pathway activation is a convergent feature in EGFR-mutant PC9 NSCLC cells with acquired resistance to EGFR TKI. They found that combining an EGFR TKI with an AKT inhibitor induced significant growth inhibition in vitro and in vivo. They also examined clinical samples and found that activated AKT was increased in the majority of EGFR-mutant patients after progression on EGFR TKIs. Moreover, the high levels of phosphorylated AKT in patients prior to EGFR TKI treatment correlate with significantly worse progression free survival (PFS) and overall survival (OS) after first-line EGFR TKI treatment. Lee and colleagues investigated possible mechanisms responsible for acquired resistance to HER2-targeted therapy in gastric cancer. They found that lapatinib (a dual EGFR and HER2 TKI)-resistant HER2-posititve gastric cancer cells upregulated phosphorylation of EGFR/HER2, and MET appeared to be closely related to the activation of PI3K/AKT and ERK1/2. Resistance to TKIs can also appear through mutations within the RTK or mutations within downstream pathways [282,284,285,287]. Given that Ras is activated downstream of these receptors, any mutations within Ras or its effector pathways will render the cell resistant to the TKI. Different studies have shown that K-RasK-Ras mutations render patients resistant to Gefitinib, which is used to treat NSCLC patients [288,289]. Similarly, treatment of colorectal cancer with anti-EGFR monoclonal antibodies cetuximab or panitumumab is only successful in a subset of patients [290] due to the appearance of Ras mutations and variations in the EGFR extracellular domain, which reduce antibody binding efficiency, initiating relapse [285]. Thus, combination treatment with PI3K inhibitors may delay the onset of resistance appearance.

Arginine (Arg) auxotrophy occurs in certain tumor types and is usually caused by the silencing of argininosuccinate synthetase 1 or arginine lyase genes. Such tumors are often associated with an intrinsic chemoresistance and thus a poor prognosis [291]. Arginine auxotrophy, however, renders these tumors vulnerable to treatment with arginine-degrading enzymes. Pegylated arginine deiminase (ADI-PEG20), which converts Arg to citrulline and ammonia, resulting in Arg deprivation, is used for the treatment of melanoma, liver cancer, and other types of cancer. ADI-PEG20 has been shown to activate Ras signaling and, therefore, ERK and PI3K/AKT/GSK-3β kinase cascades, resulting in the phosphorylation and stabilization of c-Myc by the attenuation of its ubiquitin-mediated protein degradation mechanism, which induces ADI resistance [292]. PI3K inhibitors suppress c-Myc induction and enhance ADI-mediated cell killing and in animal models of argininosuccinate synthetase (AS, the rate-limiting enzyme for arginine biosynthesis)-negative melanoma, combination therapy using a PI3K inhibitor together with ADI-PEG20 yielded additive antitumor effects as compared with either agent alone.

Many different signaling pathways and associated factors play a critical role in mediating drug resistance through Ras and PI3K pathways. Furthermore, not only can these pathways contribute to an increased cellular proliferation rate in bulk cancer, but they may also support the self-renewal and stemness properties of cancer stem cells. Whilst it seems inevitable that cancer therapy resistance will remain an issue, better targeting of its potential causes is an imperative, which requires a greater understanding of resistance mechanisms.

## 6. Ras–PI3K Signaling in the Tumor Microenvironment

Cancer cells dramatically alter their local tissue environment (TME). The clinical relevance of Ras mutations is due not only to their impact on cancer cells’ autonomous mechanisms but also to their ability to alter the way tumor cells interact and influence different components of the TME, from the extracellular matrix (ECM) to endothelial cells, cancer-associated fibroblasts (CAFs), and inflammatory/immune cells. Neo-angiogenesis, tumor inflammation, and matrix remodeling are interconnected processes that alter the expression profile of cancer cells, increasing their aggressiveness, altering their expression profile, and eventually contributing to the end-stage of tumorigenesis represented by tumor dissemination [293].

### 6.1. Ras–PI3K Interaction in Angiogenesis

Angiogenesis is essential for tumor growth and metastasis to guarantee sufficient blood supply. The TME is composed of different signaling molecules and pathways that orchestrate the intricate process of angiogenesis, and tumor cells can tilt the balance toward proangiogenic factors to stimulate vascular growth. Furthermore, the TME comprises a plethora of cell types that contribute to angiogenesis. Tumor cells exploit those cells by releasing cytokines, chemokines, and growth factors to attract them into the TME. Those recruited cell types, such as myeloid cells and fibroblasts, in turn, release their stores of proangiogenic factors to facilitate angiogenesis [294].

Ras and PI3K are expressed in both tumor cells and non-malignant cells from the TME, thereby playing a multifaceted role in the process of angiogenesis. Ras promotion of tumor-associated angiogenesis has been described a long time ago [295]. Nonetheless, evidence supporting the different mechanisms underlying Ras stimulation of endothelial cells and its role on the molecular basis of angiogenesis continue to accumulate.

The mechanisms by which Ras activation initiates and sustains pro-angiogenic processes are complex, impinge on the modulation of levels of endothelial growth factors, and also increase local inflammation and stromal remodeling. The impact of K-Ras on the regulation of the most potent angiogenesis inducer vascular endothelial growth factor (VEGF) has been extensively studied in different models [296,297,298]. Oncogenic Ras-mediated upregulation of VEGFA involves the activation of multiple signaling cascades that eventually culminate in the stabilization of HIF-a, boosting its transactivation potential at the *VEGFA* promoter [299,300,301]. Tumor cells bearing Ras mutations can promote tumor-associated angiogenesis through other mechanisms apart of VEGF secretion, such as secretion of different proinflammatory, angiogenic cytokines, such as IL-8 [302,303,304,305], IL-6, and GRO1 (also known as CXC Motif Chemokine Ligand 1, CXCL1) [306] or repression of antiangiogenic factors such as thrombospondin-1 (TSP-1) and TSP-2 [307,308]. Under hypoxic conditions, PI3K is a critical downstream effector of Ras in the induction of VEGF [309,310,311] and IL8 expression [304] to induce angiogenesis. Once produced, the proinflammatory cytokines recruit immune cells such as neutrophils and macrophages, which produce angiogenic growth factors [302,312,313].

Ras can also upregulate VEGFA via the angiogenic enzyme cyclooxygenase 2 (COX2), which through the production of prostaglandins leads to the enhancement of the cyclic AMP-dependent transcription of VEGF [295]. COX2 can increase the levels of a plethora of other endothelial growth factors, such as FGF2 and PDGF, and is required for integrin-mediated endothelial cell spreading and migration [314,315]. Inhibition of COX2 using celecoxib inhibits COX2-dependent angiogenesis via the inhibition of the PI3K/AKT/HIF-1α axis in a murine hepatocarcinoma model [316], suggesting that, at least in some cases, Ras activation of COX to induce angiogenesis might be mediated by activation of PI3K pathway.

Ras–PI3K signaling in the microenvironment is capable of orchestrating changes that promote angiogenesis and survival. Once VEGFR is activated, PI3K participate in the downstream intracellular signal transduction that induces proliferation of endothelial cells and increases vessels permeability [317,318,319]. Subcutaneous tumors formed by B16F10 or LLC1 cells in a Ras–PI3K deficient host grew more slowly than those formed in the WT host and presented a marked reduction in the amount of blood vessels [320]. The decrease in vessel density and vessel area in tumors was accompanied by an increase in necrotic areas of the tumor. The authors demonstrated that endothelial cells lacking Ras-binding to PI3K were not able to respond to VEGF or FGF-2 and concluded that intact Ras–PI3K interaction is needed in endothelial cells for new blood vessel formation induced by either VEGFA or FGF-2 and that disruption of this interaction results in deficient cellular signaling that translates into a reduced capability to form new vessels. However, since this model lacks Ras–PI3K interaction in all host tissues, additional mechanisms driven by other cellular components of the TME cannot be discarded in the global angiogenic defect. In fact, tumor-associated macrophages (TAMs) extracted from the tumors of mice lacking Ras–PI3K interaction had a reduced expression of VEGFA and TGF-β, two important angiogenic factors [320]. Soler et al. described a similar growth impairment and angiogenic defects on transplanted tumors in a mouse model heterozygous for expression of a kinase-dead *Pik3ca* in all host tissues, although this model presented an increase in the number of vessels, with reduced lumen size in the mutant [321].

### 6.2. Ras–PI3K Interaction in Tumor Immune Infiltration

Interplay between cancer cells and the immune microenvironment is important for cancer progression [322]. K-Ras influences the composition of the immune microenvironment and promotes the switch from an antitumorigenic to a protumorigenic response through different mechanisms that include the expression of several inflammatory cytokines (such as IL-6, IL-8, IL-17, and IL-22) [305,323,324], chemokines (CCL2, CXCL2, or CCL1) [325], and activation of signaling pathways such as NF-κB, which has been considered fundamental in K-Ras-induced inflammation in solid tumors. The regulation of these processes is important for the recruitment, activation, and differentiation of immune cells that eventually promote a protumorigenic environment and induce cancer cell evasion from anticancer therapies [293,326,327,328]. Although the specific role played by Ras–PI3K interaction in the regulation of these processes is not clear, PI3K controls activation of such proinflammatory pathways. For example, PI3K/AKT regulates the expression of proinflammatory cytokines such as IL-6, IL-8, IL-17, and granulocyte-macrophage colony-stimulating factor (GM-CSF) [329,330]. According to these studies, PTEN knock-down in lung epithelial cells potentiated AKT phosphorylation and enhanced production of IL-6, CXCL8, CXCL10, CCL2, and CCL5 [331]. Importantly, PI3K-driven cancer activates the NF-κB-dependent transcriptional profile, increasing expression and secretion of cytokines and chemokines, especially IL-6, which helps to generate a pro-tumor microenvironment and facilitate tumor progression [332]. Based on these results, it is possible that PI3K acts downstream of oncogenic Ras to promote a proinflammatory microenvironment.

K-Ras tumors are usually infiltrated by immune cells. CD8^+^ T lymphocytes and natural killer cells (NK) generally have antitumor functions, whereas regulatory T-cells (Tregs), myeloid-derived suppressor cells (MDSCs), neutrophils, and macrophages are usually pro-tumorigenic [327]. The presence of TAMs is generally associated with tumor progression, and this is also valid in K-Ras-induced lung tumors [325,333]. Although the molecular mechanism is unknown, different studies have established that Ras signaling to PI3K may regulate macrophage infiltration in cancer. Disruption of Ras binding to PI3K, either constitutively or just in host tissues, severely impairs the recruitment of F4/80+ macrophages to tumors in K-Ras-driven lung tumors [164,320], and the few TAMs present in the tumors are closer to an M1 phenotype, which is associated with good prognosis in context of cancer [320]. Macrophage reliance on PI3K signaling for tumor recruitment is not limited to the p110α isoform. PI3Kδ-KO macrophages transferred to tumor-bearing NSG mice (NOD scid γ mouse) reduced tumor growth compared to the WT macrophages [334]. Furthermore, the blockade of PI3Kγ in pancreatic ductal adenocarcinoma (PDAC)-bearing mice reprograms TAMs to stimulate CD8^+^ T-cell-mediated tumor suppression and to inhibit tumor cell invasion, metastasis, and desmoplasia [335]. The importance of PI3Kγ for protumorigenic macrophages function was revealed by Kaneda et al., who showed that PDAC tumor growth and metastasis depend on macrophage PI3Kγ [335].

PI3K has been found to mediate immune evasion by downregulating major histocompatibility complex Class I (MHCI) antigen presentation at the cell surface, therefore decreasing cancer cell recognition by CD8^+^ cytotoxic T-cells [336,337]. Activation of PI3K signaling repressed the induction of MHCI and MHCII in oral carcinoma cells [338], and PI3K inhibition improved MHCI presentation and rendered tumor cells sensitive to recognition by CD8^+^ T-cells [339]. This effect may be, at least partially, regulated by Ras activation of PI3K, since the inactivation of oncogenic Ras increased MHCI expression in human colorectal cell lines [340], and a murine model of oncogenic Ras has been associated with the downregulation of MHCI surface expression [341]. Another mechanism by which Ras participates in immune escape is the stimulation of Treg development [342,343]. Tregs may be required for K-Ras-mediated lung tumorigenesis [344], and oncogenic K-Ras increases Treg production through the induction of the expression of IL10 and TGFβ1 after the activation of the MEK–ERK–AP-1 pathway [345]. Although the PI3K pathway was not directly evaluated, considering that TGFβ is the main cytokine implicated in the Treg induction phenotype [346] and that PI3K plays a pivotal role in regulation of the TGF-β1 expression [347,348,349], it is likely that PI3K signaling downstream of Ras participates in Treg cell generation. It is also important to consider that the Ras–PI3K pathway in non-oncogenic cells could modulate Tregs, as PI3K/AKT signals are necessary for Treg development and function [350]. As an example, PI3Kδ genetic ablation or pharmacological inhibition reduce Treg infiltration in preclinical mouse tumors and peripheral tissues [351,352], while PI3Kα/δ inhibition directly promotes durable anti-tumor responses and enhances cytotoxic T-cell function in vivo [353].

Neutrophils are also described as key drivers of cancer progression [354]. One of the mechanisms by which tumor-associated neutrophils (TANs) promote tumor progression is through the generation of ROS that mediates the inhibition of T-cell proliferation, creating an immunosuppressive environment [355]. Analysis of data from uterine corpus endometrial carcinoma (UCEC) patients showed that high expression levels of *Pik3ca* correlate with the neutrophil-related pathway Mac1 Integrin signaling, which is the most abundant integrin on neutrophils and is substantially up-regulated on the cell surface upon neutrophil activation [356]. Consistently, neutrophil-related genes and neutrophils were significantly altered by *Pik3ca* expression in UCEC [357].

From a clinical point of view, targeting the immune microenvironment is a promising area, and several therapies are currently used in clinical practice, such as antibodies targeting immune-checkpoint molecules (e.g., programmed cell death 1 (PD-1), programmed cell death ligand 1 (PD-L1), and cytotoxic T-lymphocyte-associated protein 4 (CTLA-4)) [328,358]. However, there are tumors that do not respond to immunotherapy, as is the case for pancreatic cancer [359], and NSCLC patients do not respond to anti-CTLA4 therapy [360]. Thus, there is an urgent need to understand why some patients do not benefit from immunotherapy. The success of the immune-checkpoint blockade depends on the immunogenicity of the tumor, so it is critical to understand the molecular mechanisms that dictate tumor cell PD-L1 expression. K-Ras mutations in lung cancer have been associated with increased PD-L1 expression causing an improved clinical response to anti-PD1 therapy [328], although the mechanism by which oncogenic K-Ras augment PD-L1 expression is controversial. Some studies indicate that PD-L1 is up-regulated by K-Ras mutation through p-ERK but not p-AKT signaling [361,362]. However, Coelho and colleagues showed that oncogenic Ras signaling regulates PD-L1 expression through both MEK and PI3K pathways, with MEK pathway activation stabilizing PD-L1 mRNA through the inhibition of tristetetrapolin (TTP) and the PI3K pathway only affecting protein expression [363]. In agreement with these data, K-Ras-dependent activation of PI3K/AKT/mTOR in human lung adenocarcinma and squamous cell carcinomas tightly regulate PD-L1 expression both in vitro and in vivo. This is further supported by studies carried out on patient samples, suggesting that oncogenic K-Ras can cause an immune escape by AKT/mTOR pathway via PD-L1 [364]. Importantly, PI3K has been shown to regulate PD-L1 immunosuppressive function even in the absence of oncogenic Ras. MDSCs activate the PI3K/AKT/NF-κB pathway in B-cells via the PD-1/PD-L1 axis and drive immunosuppressive effects in breast cancer [365]. K-Ras activation in pancreatic cancer is also associated with increased PD-1 expression, although immunotherapy has limited clinical success in this indication [328,359]. The abundance and complexity of the TME in pancreatic cancer may explain the lack of effectiveness of conventional immunotherapy treatments. Thus, combinatorial strategies targeting the immune system (e.g., PD-L1) and the TME complexity (e.g., with inhibitors of colony stimulating factor receptor 1(CSFR1) or chemokine C-X-C receptor 4 (CXCR4)) are in clinical trials, and it is expected that the reprograming of the TME will improve the benefit of classical immunotherapy treatment [328,359].

### 6.3. Ras–PI3K in CAFs and ECM Remodeling

CAFs are one of the most abundant stromal cell types in the tumor microenvironment and contribute to several aspects of the malignant phenotype [366]. The relevance of Ras and PI3K signaling in CAFs is, by far, less understood than in cancer cells, but different studies have validated the role played by PI3K in the protumoral functions of CAFs. For example, CAF-derived TGF-α promotes the metastatic colonization of ovarian cancer cells via activation of AKT, epidermal growth factor receptor (EGFR), and extracellular signal-regulated kinase (ERK)-1/2 signaling pathways [367]. Activation of PI3K due to loss of PTEN in stromal fibroblasts accelerated initiation, progression, and malignant transformation of mammary epithelial tumors due to the induction of genes involved in ECM remodeling and the recruitment of macrophages, including *Mmp9* and *Ccl3* [368]. PI3Kγ has been shown to regulate TNF-mediated secretion of matrix metalloproteinases (MMPs) from fibroblasts, which is crucial for the migration of tumor cells [369]. Ziqian Li et al. reported that in colorectal carcinoma and melanoma, CXCL5 derived from the CAFs activated PI3K/AKT signaling pathway and promoted the expression of PD-L1 in tumor cells [370].

CAFs constitute a heterogeneous population of cells, and their origin is still under debate. The most widely accepted view is that CAFs are early developmental precursors of cells with different origins that respond to signals derived from cancer cells [371]. Among these signals, some are Ras–PI3K-mediated, such as TGF-β, HGF, EGF, and FGF2 [372]. Thus, it is plausible that Ras–PI3K signaling has a prominent role in CAFs expansion and function and that its inhibition may influence tumor growth and progression, although additional studies addressing the relevance of this signaling pathway in CAF functionality are required.

Modulation of cancer-associated fibroblasts by K-Ras mutational changes through Hedgehog signaling (Hh) has been suggested as an important mediator of CAF activation in PDAC models [373,374,375,376,377]. SHh originated from cancer cells was responsible for changes on the fibroblast proteome, promoting the synthesis of ECM components, such as collagen, and matrix metalloproteinases (MMPs) [376]. Cancer-derived Hh also promoted the expression of growth factors such as IGF1 and GAS6 by fibroblasts, which reflected back on cancer cells. Additionally, fibroblast-derived signals significantly alter the phosphoproteome of PDAC cells, not only promoting an increase in the usual cell-autonomous alterations but also activating pathways that are not autonomously triggered. Recently, it was shown that the conditioned medium of K-RasV12-overexpressing colorectal cells strongly increased the migration of intestinal fibroblasts without affecting their proliferation rate and their differentiation status [378]. Together, these studies suggest that oncogenic Ras in cancer cells can alter the behavior of fibroblasts, which in turn affects the microenvironment through ECM changes and growth factor signaling, contributing to tumor progression.

The ECM is mainly secreted by CAFs in the TME and provides structural scaffolding for the surrounding cells as well as biochemical and biomechanical cues for cell differentiation, proliferation, and migration. Within the tumor stroma, not only the cancer cells but also the resident fibroblasts, which differentiate into CAFs, modify the ECM. Growth factors and chemokines, which are tethered to and released from the ECM, as well as metabolic changes of the cells within the tumor bulk, add to the tumor-supporting tumor microenvironment. CAF-mediated ECM remodeling is a highly responsive process of receiving, processing, and responding to the cellular, molecular, and mechanical signals in the TME. The lysyl oxidase (LOX) family and MMPs represent two major types of remodeling enzymes produced by CAFs with a high relevance in tumor progression. As a highly adaptive and mechanically responsive stromal cell type, CAFs sense and respond to the ECM stiffness in a LOX/MMP-dependent manner and further fine-tune the CAF-ECM interactions [379]. LOX is a secreted amine oxidase that modifies the primary tumor microenvironment by crosslinking collagen and elastin in the ECM [380,381] and has been linked to metastasis of CRC cells such as SW480 and its patient-matched metastatic clone, SW620 [382,383]. LOX expression is enhanced by co-expression of Ras G12V through the activation of PI3K/AKT and concomitant accumulation of the hypoxia-inducible factor (HIF)-1α [384,385]. Treatment of cancer cells carrying activated Ras with Kobe0065 [386] (a small-molecule inhibitor of Ras) or a siRNA targeting Ras downregulates the expression of LOX, which has been implicated in metastasis. Moreover, Kobe0065 effectively inhibits not only migration and invasion of cancer cells carrying the activated *ras* genes but also lung metastasis SW620 carrying the K-Ras G12V mutation. Similarly, in a murine model of pancreatic cancer driven by oncogenic Ras (KPC mice), the targeting of LOX leads to a remarkable increase in survival, especially when combined with gemcitabine. LOX inhibition was associated with increased tumor vascularization, immune cell infiltration, and drug efficacy [387].

Plasticity of the ECM is exploited by tumors during the multi-step process, leading to the acquisition of more aggressive and invasive features [388]. ECM degradation promotes tissue invasion by removing the physical barriers that contain the tumor and prevents its migration towards vessels. The MMP family of proteases degrades ECM proteins and is often deregulated in cancer. Increasing evidence links PI3K/AKT signaling to the control of MMP production in normal and cancer cells [332,369,389,390,391,392,393,394], which is not entirely surprising considering that factors known to be regulated by Ras–PI3K, such as growth factors (TGF-β, EGF, PDGF and FGF-2) and inflammatory cytokines (IL-1, IL-6 and TNF-α), are regulators of MMP levels [395]. Most of our current knowledge on PI3K regulation of CAFs comes from tumor cells, but there is a concerning lack of understanding of how PI3K activation in CAFs influenced by tumor cells regulates MMPs to drive ECM remodeling and tumor cell spread.

A direct role of mutant K-Ras in the modulation of ECM properties has also been described. In K-Ras G12V-driven lung cancer, epithelial cells secreted higher levels of activated MMP-9 [396], and in pancreatic cancer mouse models, increased expression of MMP3 cooperates with K-Ras activation to shape the stromal microenvironment, not only by stimulating immune cell influx but also as a primary proteolytic activator of MMP-9 [397]. Additionally, in pancreatic cancer cells, mutational activation of K-Ras induced the expression of the eukaryotic translation initiation factor 5A (eIF5A) and consequent stimulation of ROCK1 and ROCK2 [398]. ROCK activation and signaling ultimately results in ECM remodeling and collagen degradation by MMPs, thereby enabling invasive tumor growth through elimination of physical restraints [399].

It is now evident that Ras signaling not only exerts its activity to drive cell growth and sustain survival in cancer cells, but also modulates microenvironmental changes. In stromal cells, Ras signaling contributes to directing the response to a multitude of cues that cancer cells produce to modify their microenvironment. It is, thus, an important area to explore, not only from basic knowledge, but also from a therapeutic point of view.

## 7. Targeting Ras and PI3K in Cancer

K-Ras is a major clinical target, as it is by far the most significant form of RAS in terms of cancer incidence. Regardless of the tremendous attempts in the past decades to develop inhibitors, Ras was considered undruggable for decades [400]. Recently, several small molecules (AMG510, MRTX849, JNJ-74699157, and LY3499446) have been developed to specifically target K-Ras G12C, culminating with the FDA approval of Sotorasiv (AMG510) [401]. Further discussion of the different approaches used over the past decades to inhibit oncogenic Ras is provided by a recent review from Molina-Arcas et al. [402].

Hyperactivation of the PI3K pathway in cancer, and its vital function in cell survival and proliferation, have made it an ideal target for treatment. Furthermore, the inhibition of Ras-effector pathways has for years been considered the most effective approach to target Ras-driven tumors [402,403]. Results from clinical trials with PI3K inhibitors in solid tumors have been, however, largely disappointing [404]. Reasons for this failure include drug resistance, aberrant activation of parallel signaling pathways that in turn stimulate the PI3K pathway, and intolerable toxicity due to lack of specificity, especially with pan-PI3K inhibitors [74,405], while isoform-specific inhibitors seem to have better therapeutic efficacy and an improved toxicity profile [406]. So far, only five PI3K inhibitors have been approved to use in clinical practice. One of them is Alpelisib, a PI3Kα inhibitor approved for breast cancer treatment and investigated in combination with a range of targeted therapies and chemotherapies in multiple clinical trials [407]. The other PI3K inhibitors are used in the treatment of hematological cancers: idealisib, a PI3Kδ inhibitors [404], copanslisib, the only pan-PI3K inhibitor approved [408], duvelisib, a dual inhibitor of the PI3Kδ and PI3Kγ isoforms [409], and umbralisib, a dual inhibitor of PI3Kδ/casein kinase-1εr (www.fda.gov, (accessed on 20 May 2021) [410]).

Considerable efforts have been made to improve the clinical efficacy of PI3K inhibition and to reduce adverse effects. The generation of isoform-selective PI3K inhibitors may reduce the intrinsic toxicity associated with pan-PI3K inhibition, allowing the exploration of combination therapies. In line with the evidence for isoform-specific PI3K targeting in preclinical models, pivotal clinical trials indicate new opportunities for p110α-selective inhibition in selected patient populations. Results from clinical trials with taselisib show that this p110α specific inhibitor can more effectively suppress the PI3K signaling pathway, resulting in greater anti-tumor activity and an improved therapeutic index [411,412]. Since different PI3K isoforms have different expression patterns and activation mechanisms, inhibition of specific PI3K isoforms involved in specific tumors appears to be a way to circumvent toxicity issues. An example of success is the PI3Kδ-specific inhibitor idelalisib, approved for the treatment of B-cell lymphoid malignancies [413]. These promising effects are likely due to the concomitant action of the drug in both tumor and TME.

The emerging roles of Ras and PI3K in shaping the tumor microenvironment provide a new awareness of how kinase inhibitors may contribute to enhancing antitumor immunity and their application in combination therapy to improve the outcome of immunotherapy. For example, PI3Kα or PI3Kβ inhibitors impair tumor cell growth and induce changes in the tumor microvasculature, which correlates with antitumor activity and, more importantly, potentiates the anticancer activity of current cytotoxic therapy [319,321,414,415]. Furthermore, PI3Kβ inhibition could improve the efficacy of immunotherapy by increasing T-cell infiltration in melanoma with PTEN loss [416]. Additionally, in leukemic B-cells, PI3Kδ blockade impairs pro-survival signaling and induces tumor cell death. On the other hand, the inhibition of PI3Kδ in monocyte-derived cells of the lymph node stroma blocks the secretion of pro-survival chemokines sensed by tumor cells, potentiating the pro-apoptotic effect of the drug [417].

The same principle could be applied to PI3Kγ and PI3Kδ specific inhibitors used for the treatment of solid tumors. PI3Kδ inactivation protects against the development of a broad range of cancers, which is mainly associated with the disruption of the function of Tregs and possibly of MDSCs [351]. The anticancer activity of PI3Kγ seems to be associated with its effects on tumor macrophages, which translates into the stimulation of T-cell recruitment and reduced tumor growth [335]. Important results with PI3Kδ and PI3Kγ inhibitors, used alone or in combination with standard anticancer therapy, have been obtained in murine models of pancreatic cancer, a tumor that is highly resistant to chemotherapy and radiation therapy and does not respond to immune checkpoint inhibitors [98,335,351,418,419]. For example, the deletion of PI3Kγ improved survival and responsiveness to standard-of-care chemotherapy in animal models of PDAC [335], and the dual inhibition of PI3Kγ and CSF-1/CSF-1R reduced the tumor infiltration of MDSCs and inhibited tumor growth [420]. Importantly, PI3Kγ inhibition can synergize with anti-PD-1 to reduce tumor growth, extending survival [421,422], and pan-PI3K inhibition can be used to overcome resistance to immune checkpoint inhibitors [423].

As we expand our understanding of the role of PI3K inhibitors in anticancer and procancer immunity, concerns are raised over the implication of isoform-unspecific PI3K inhibitors in anticancer immune response. Although most PI3K inhibitors show a boosting effect on anticancer immunity, it is anticipated that selective PI3K inhibitors can maximally enhance host immunity against cancer and minimize deleterious effects on normal tissues.

## 8. Conclusions

Signaling networks that are triggered by oncogenic Ras within the cell are complex and highly dynamic. Enormous efforts have been made to elucidate the role of different effector pathways in Ras-controlled functions. In the last few years, our capacity to study the in vivo ramifications of the expression of oncogenic Ras has been constantly improving due to the development of sophisticated genetically engineered mouse models that feature activating mutations of Ras. By achieving tissue- and cell-specific expression in a time-controlled and reversible manner, these models often recapitulate the genetic and biological evolution of human cancers, increasing our understanding of the crucial mediators of RAS-driven oncogenesis and also being instrumental to testing and developing novel targeting strategies directed at Ras.

We now know that activation of K-Ras on cancer cells reshapes the tumor microenvironment, both by oncogenic signaling coming from tumor cells affecting other cells of the stroma, but also by normal signaling within stromal cells. This effect will, in turn, affect tumor cells’ behavior. A complete characterization of the paracrine effects of K-Ras-mutant cancer cells in models with mutant K-Ras would be valuable to obtain an overall view of these effects, identify tumor specificities, and point to possible combination therapies with stromal modulatory approaches. Since metastatic outgrowth relies on the recruitment of non-cancer cells, such as myeloid cells, endothelial cells, fibroblasts, and ECM remodeling, a better understanding of the paracrine effect of Ras signaling would help address the specific role played by mutant K-Ras cancer cells in the regulation of these stromal components and how this would impact on the establishment of metastatic lesions. Additionally, because different K-Ras mutations have distinct transforming potential and preferentially activate different effector pathways, it would be relevant to understand whether they impact the interaction between cancer cells and the microenvironment differently and which signaling pathways have a prevalent role in each process. This would be relevant for the development of mutation-specific therapeutic approaches.

Considering that cancer cell communication with the microenvironment is dynamic and not a unidirectional process and that mutual regulation of cancer cells and ECM exists, understanding the role of Ras in the integration of external signaling and the subsequent cancer cell response will be essential to identifying key molecules that mediate this cross-talk and will undoubtedly impact the design of new combinatorial therapeutic strategies aimed at targeting tumor cells:microenvironment dependencies.

## Figures and Tables

**Figure 1 genes-12-01094-f001:**
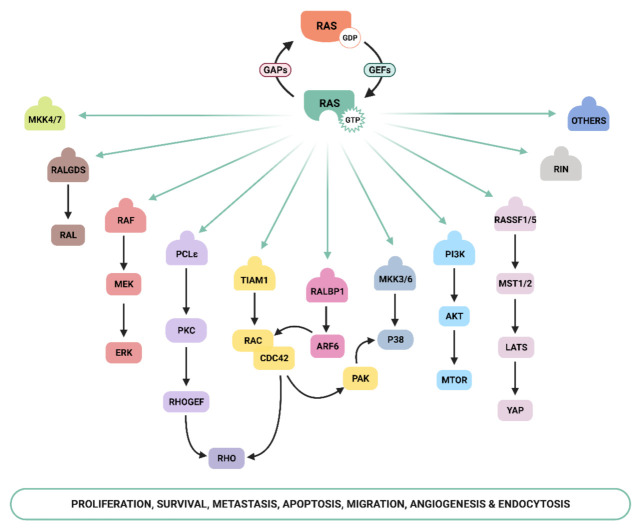
Ras effector pathways. GTP binding to Ras induces changes in protein conformation that increases Ras affinity for downstream effectors, thus activating many different signal transduction pathways. Among them, Raf/MEK/ERK, PI3K/AKT, and Ral are the best understood. Further, Ras crosslinks with other signaling pathways such as proteins from the Rho family and YAP. Through the regulation of the different downstream pathways, Ras proteins play a key role in the control of proliferation, survival, metastasis, apoptosis, migration, angiogenesis, and endocytosis.

**Figure 2 genes-12-01094-f002:**
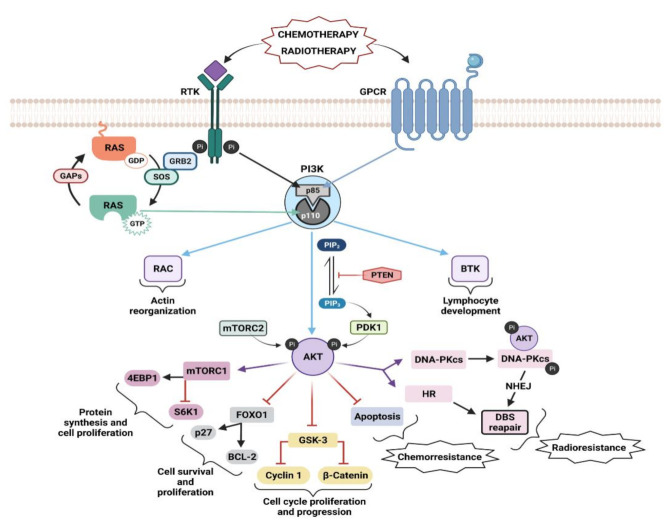
Mechanisms of activation of PI3K and downstream effectors. GCPRs and RTKs are upstream signals that control PI3K activation through direct interaction with the regulatory subunit of PI3K. Further, RTK can activate PI3K indirectly through Ras activation that in turn activates PI3K in a p110-dependent manner. Once activated, PI3K generates PIP_3_ that promotes AKT phosphorylation, which subsequently phosphorylates a large number of downstream targets to control cell survival, proliferation and apoptosis. Other PI3K effectors are TEC family tyrosine kinase, such as BTK, and GTPases of the Rho/Rac/cdc42 family. Activation of PI3K-AKT pathway is an important mechanism in the development of resistance to chemotherapy (through protection of drug-induced apoptosis [79]) and radiotherapy (through repair of radiation-induced DSBs [80,81]).

**Figure 3 genes-12-01094-f003:**
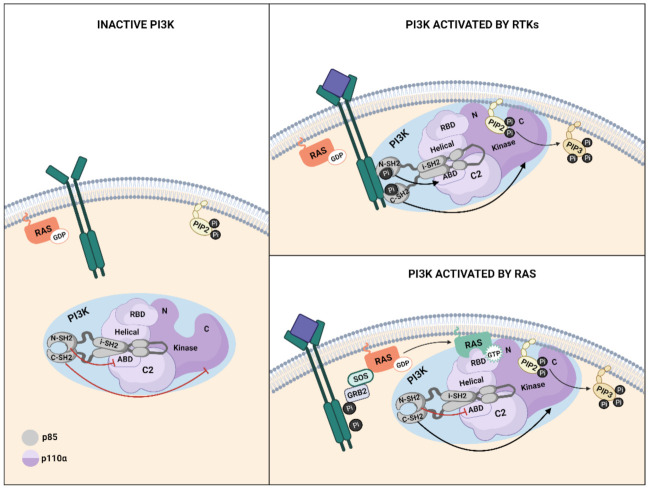
PI3K activation. Under unstimulated conditions, interactions between the regulatory and the catalytic subunit keep PI3Ks in the inactive state. The nSH2 and iSH2 (coiled portion) domains of p85 are the minimal fragment of the regulatory subunit required for full inhibition of p110α lipid kinase activity. In PI3K activation by RTKs, binding of p85 to phosphorylated RTKs disrupts the inhibitory contact between the regulatory and the catalytic subunit, which generates conformational changes in p110α for substrate catalysis. In PI3K activation by Ras, the interaction of Ras with p110 displaces nSH2 and iSH2 in p85 from the p110α subunit, facilitating the full activation of PI3Kα.

## Data Availability

Not applicable.

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
