# Peer review of "The Importance of Being PI3K in the RAS Signaling Network"

_genes, 2021, doi:10.3390/genes12071094_

Round 1
Reviewer 1 Report
This is a review that cover a large amount of litterature. The topic is relevant for the journal and is very important.
I have only a few points to discuss:
- while I understand that the main interest of the authors is lung cancer, I think it is important to discuss the context of pancreatic cancer where genetic clear-cut experiments showed that PI3K signalling overuled MAPK pathway activation, 2 labs worked on this topic PMID: 25452273 + PMID: 23453624
- l 131 to 141 - because this is a general review, in my opinion, it would be important to cite the work that show in vivo importance of direct coupling of class IA PI3Ks to RTKs instead of ref 25 (work citing KO of p85s for example) as well as direct coupling of p110gamma and p110beta to Gbg (work of the first KO of p110gamma PMID: 10669418 and work of the first KI of p110beta that dissect this point PMID: 18544649) instead of reviews, then cite the reviews for further details
- Figure 2 has an issue as it does not place activation of PI3Ks by GPCRs correctly - PI3Ka is not directly coupled to GPCRs
- Figure 3 - describe co-stimulation as well
- line 275 I am not convinced by the references - there are some seminal work that combine genetic and pharmacological inactivation of PI3K isoforms and Kras mutation that are not cited or referenced here (such as PMID: 24569456 ) - also details on isoform specific inhibitors should be mentioned – a specific paragraph on isoform selective roles of PI3Ks should be added as it is instrumental to the understanding of Ras-PI3K signalling
Minor details :
- mention that Drosophila does have isoforms of class I PI3Ks
- paragraphs 4.1 and 4.2 are very long, sub-title should be proposed
Author Response
This is a review that cover a large amount of literature. The topic is relevant for the journal and is very important.
We thank the reviewer for this positive comments.
I have only a few points to discuss:
- while I understand that the main interest of the authors is lung cancer, I think it is important to discuss the context of pancreatic cancer where genetic clear-cut experiments showed that PI3K signalling overuled MAPK pathway activation, 2 labs worked on this topic PMID: 25452273 + PMID: 23453624.
We thank the reviewer for this comment. We have now added a new paragraph describing the importance of RAS-PI3K signalling in pancreatic cancer (lines 506-513): “p110α is also critical for K-Ras-driven pancreas carcinogenesis as p110α inactivation prevents mouse lethality and the appearance of all types of pancreatic lesions induced by mutated K-Ras [170]. Supporting these data, Eser et al demonstrated that oncogenic Pi3kca mouse models show similar patterns of acinar-to-ductal metaplasia (ADM), pancreas cancer progression and similar activation of key downstream effectors of PI3K that the K-Ras G12D model, indicating that PI3K signals downstream of mutated K-Ras in pancreatic cancer [171]. Further, PDK1, AKT or PI3K inhibition resulted in normal life expectancy and inhibition of pancreas cancer development in the K-Ras G12D model.”
- l 131 to 141 - because this is a general review, in my opinion, it would be important to cite the work that show in vivo importance of direct coupling of class IA PI3Ks to RTKs instead of ref 25 (work citing KO of p85s for example) as well as direct coupling of p110gamma and p110beta to Gbg (work of the first KO of p110gamma PMID: 10669418 and work of the first KI of p110beta that dissect this point PMID: 18544649) instead of reviews, then cite the reviews for further details.
We have now added the references that the reviewer suggested to lines 131-141.
- Figure 2 has an issue as it does not place activation of PI3Ks by GPCRs correctly - PI3Ka is not directly coupled to GPCRs.
We apologise for the mistake. Figure 2 has been corrected.
- Figure 3 - describe co-stimulation as well.
We think reviewer refers to co-stimulation of PI3K activation by p85 once RAS is bound to p110a. Although it is an important point on RAS-dependent activation of PI3K, we wanted to present a clear picture of the differences in the residues participating in RTK-mediated PI3K activation or Ras.
- line 275 I am not convinced by the references - there are some seminal work that combine genetic and pharmacological inactivation of PI3K isoforms and Kras mutation that are not cited or referenced here (such as PMID: 24569456) - also details on isoform specific inhibitors should be mentioned – a specific paragraph on isoform selective roles of PI3Ks should be added as it is instrumental to the understanding of Ras-PI3K signalling.
We thank reviewer for this comment. We have now added PMID: 24569456. Regarding to the mentioning of isoform specific inhibitors, the scope of this review is to understand the role of this effector pathway in the Ras context, more than a review on how different PI3K isoforms interact with Ras. As reviewer mention, it is an important point and we have discussed in the different sections when a particular Ras-related function is linked to the activation of a specific PI3K isoform. Thus, addition of a new paragraph/section based on isoform selective roles of PI3K would involve reformatting of the whole review to avoid duplication.
Minor details:
- mention that Drosophila does have isoforms of class I PI3Ks.
We thank reviewer for this suggestion. We have mentioned Drosophila through the text to refer to some particular experiments performed to identify the relevance of Ras-p110a binding. We have not conducted a comparison of the different PI3K family members found in different organisms in this review and consider that if we mentioned it for Drosophila, other organisms should also be mentioned. We think it is not the main scope of this review as currently written.
- paragraphs 4.1 and 4.2 are very long, sub-title should be proposed
We thank the reviewer for this suggestion. Sections 4.1 and 4.2 are of similar length than others present in the review, that have not been subdivided. We pretend to provide a general overview of Ras-PI3K interaction in cell growth and apoptosis (section 4.1) since in many instances both functions are linked and the same studies have revea
Reviewer 2 Report
The manuscript by Cuesta et al is a fairly comprehensive review of PI3K signaling. It is well organized and well written. There are two topics important to the role of PI3K signaling, specific to Ras-mutated cancers, that should be included in the review.
- Rebound PI3K activation is an important contributor to resistance to MEK and KRASG12C inhibitors. This is not covered in the review.
- In some KRAS-mutated cancers PI3K signaling is independent of mutant KRAS. This has important therapeutic implications. Ebi et a 2011 JCI and Molinas-Arcas 2013 Cancer Discovery should be referenced.
- In the text describing Fig. 2 (pgs 4-6), I did not see references to activation of DNA-PKs or HR for DBS repair. This comes much later in the text but should be mentioned here.
Author Response
The manuscript by Cuesta et al is a fairly comprehensive review of PI3K signaling. It is well organized and well written.
We thank reviewer for this positive comments.
There are two topics important to the role of PI3K signaling, specific to Ras-mutated cancers, that should be included in the review.
- Rebound PI3K activation is an important contributor to resistance to MEK and KRASG12C inhibitors. This is not covered in the review. LÍNEAS 707-732
We thank reviewer for this suggestion. We have now added two paragraphs (lines 696 to 716) discussing both aspects of therapy resistance.
“ERK activation is a common feature of tumors with K-Ras, N-Ras, or B-Raf mutation [268], and inhibition of Raf/MEK/ERK pathway was supposed to be effective in cancers har-bouring mutations in these genes [269]. However, a portion of patients developed drug re-sistance mechanisms and no longer responded to Raf or MEK inhibitors [270]. Upregula-tion of PI3K pathway was found to be a major mechanism of resistance to Raf and MEK inhibitors [271-273]. The activation of PI3K after ERK pathway inhibition comes from dif-ferent mechanism that include RTK reactivation [273,274], activating mutations in Pi3kca or loss of PTEN [275] or activation of positive feedback loop composed of GAB1, Ras, and PI3K, which induces the bypass of the ERK signal to the PI3K signal [276].
Recently, small molecules against K-Ras G12C mutations have been developed and the first drug targeting this mutation has been approved by FDA {Mullard, 2021 #138}. How-ever, therapeutic resistance to K-Ras G12C inhibition has been observed in preclinical tu-mor models and also in the clinic [277-279]. PI3K pathway may be implicated in the re-sistance to K-Ras G12C inhibitors [280] as preclinical studies have shown failure to inac-tivate PI3K signaling pathway after treatment with G12C inhibitors [278,281,282]. Im-portantly, combination of G12C and PI3K pathway inhibitors was effective in vitro and in vivo on models that are resistant to single-agent G12C inhibitor [278], or significantly im-proved antitumor activity of G12C inhibitors [281,283], what could be explained by a concomitant inhibition of p-ERK (due to G12C inhibition) and p-AKT [278,281]. Such combination could avoid the toxicity associated with inhibition of MEK and PI3K, while efficacy of inhibiting both pathways in tumor cells is maintained [283).
- In some KRAS-mutated cancers PI3K signaling is independent of mutant KRAS. This has important therapeutic implications. Ebi et a 2011 JCI and Molinas-Arcas 2013 Cancer Discovery should be referenced. LÍNEAS 495-509
We thank reviewer for this suggestion. We have now added some text discussing the role of PI3K in mutant and WT RAS tumors (lines 491-504): “According to the observation that PI3K mediates oncogenic signaling in Ras-WT cancers, Molina-Arcas and colleagues showed that PI3K is critical for survival of both K-Ras mutant and K-Ras WT NSCLC cells, as PI3K inhibition induced loss of cell viability irrespective of the genotype [169]. However, opposite results on the role of oncogenic Ras on PI3K activation were obtained in colorectal cancer cells carrying mutations in K-Ras, where knockdown of K-Ras did not suppress AKT phosphorylation and PI3K/AKT pathway required insulin-like growth factor I receptor (IGF-IR)-induced activation. This is therapeutically important as, while suppression of mutant K-Ras is not sufficient to downregulate PI3K/AKT, IGF-IR inhibition could suppress PI3K signaling in K-Ras mutated colorectal cancers [170]. In summary, PI3K is not under the sole control of mutant KRAS and consequently, to discern the mechanism of PI3K activation in different cancers seems to be essential for the development of new drugs”.
- In the text describing Fig. 2 (pgs 4-6), I did not see references to activation of DNA-PKs or HR for DBS repair. This comes much later in the text but should be mentioned here.
We apologize for this mistake. We have now added explanation and references to figure 2 legend
